# Association between Family Level Influences and Caries Prevention Views and Practices of School Children in a Sub-Urban Nigerian Community

**Abiola Adetokunbo Adeniyi** [1,*]**, Morenike Oluwatoyin Folayan** [2,3,4,5]**, Olaniyi Arowolo** [6]**, Nneka Maureen Chukwumah** [7] **and Maha El Tantawi** [8]

1   Department of Oral Health Sciences, Faculty of Dentistry, University of British Columbia, Vancouver, BC V6T 1Z3, Canada
2   Department of Child Dental Health, Obafemi Awolowo University, Ile-Ife 220282, Nigeria
3   Nigeria Institute of Medical Research, Yaba 101212, Nigeria
4   Community Oral Health Department, Tehran University of Medical Sciences, Teheran 1449614535, Iran
5   Faculty of Health Sciences, University of Zaragoza, 50009 Zaragoza, Spain
6   Department of Child Dental Health, Obafemi Awolowo University Teaching Hospitals Complex, Ile-Ife 220282, Nigeria
7   Department of Preventive Dentistry, School of Dentistry, College of Medical Sciences, University of Benin, Benin City 300271, Nigeria
8   Department of Preventive Dentistry, Alexandria University, Alexandria 21527, Egypt
*   Correspondence: abiola71@dentistry.ubc.ca

**Abstract:** Little is known about how family-level factors influence children's caries prevention views and practices in Nigeria. The purpose of this study was to assess the associations between family level characteristics and caries prevention views and practices of 6–11-year-old primary school children. Data was collected through a cross-sectional survey of 1326 children in Ile-Ife, a Nigerian suburb. The child's family structure, size, and birth rank were independent variables while the child's caries prevention views and self-care practices were dependent variables. Multivariable logistic regression analysis was conducted to identify risk indicator(s) for caries prevention views and practices. The study participants' mean (SD) age was 8.7 (1.9) years, 407 (30.7%) children had positive caries prevention views, and 106 (8.0%) children did not use the recommended self-care caries preventive methods. Children from larger families had significantly lower odds of having positive prevention views (AOR: 0.572; *p* = 0.002). Children who were not living with both parents had higher odds of using recommended self-care caries preventive methods (AOR: 3.165; *p* = 0.048). The findings suggest that family size and family living structure may be social determinants of caries risks in children 6–11 years old in the study population. These findings need to be studied further.

**Keywords:** household factors; caries; knowledge and practice; school-aged children

## 1. Introduction

The World Health Organization emphasised the need for expanded access to preventive dental care at the population level [1] in its 2021 resolution. The Organization also called for formulation of community-based policies and legislation to promote oral health throughout the life span [1]. The resolution focused on dental caries since it is the most common chronic disease of childhood [2,3], has a high treatment cost, has a major impact on quality of life [3], and is preventable. Caries prevention strategies should encourage the use of preventive practices that positively change the oral bacterial flora, modify diet, increase tooth acid resistance, and/or reverse tooth demineralization [4–6]. However, individuals can adopt oral health promoting practices only if they have a positive view towards oral disease prevention [7,8]. Despite the low prevalence of caries among children in Nigeria [5],

the high levels of untreated caries and the limited access to oral health care services [9] make it vital to promote the adoption of preventive practices.

There is very little known about the impact of school children's views about caries prevention on caries risk and experience in Nigeria. What is, however, known is that the knowledge of children in Nigeria about caries prevention is poor [10,11], and it is influenced by socio-economic factors [10,11] and parental knowledge of caries prevention [11]. These highlight the role of family level factors in the risk of children to caries.

Families directly and indirectly influence children's oral health through giving direct support and through role modelling [12]. The Fisher-Owens model, a multilevel conceptual framework that recognizes the complex interactions between individual, family-level, and community-level factors and the five health determinants domains, recognizes the role of family influence on health behaviors. It also recognizes the impact of the complex interactions between the multiple levels of influences on the child's oral health [12].

Family-level influences on children's oral health behavior are predominantly driven by mothers [13,14], although older siblings wield some influence [15]. Children's caries experience is significantly worse when family functioning is poor [16,17] and parenting styles are authoritative or permissive [18]. Family level factors associated with caries risk of children resident in Nigeria include birth rank [19], family size [20], and family structure [20]. However, very little is known about the broader family-level influences on children's perspectives of caries prevention: children's views of caries prevention is a very important caries protective factor [7]. Understanding the impact of family context on the views of children about caries prevention may influence oral health promotion policy formulation and preventive oral health programming for school-aged children in Nigeria. Identifying factors that can inform the design and implementation of school-based oral health programmes for school-aged children is a strategic approach for promoting equitable access to oral health care [21,22].

The primary objective of this study was therefore to assess the views of school children aged 6–11-years in Ile-Ife, a sub-urban community in Nigeria, on dental caries prevention, as well as to determine the associations between the child's views about caries prevention and practices their practices with family level factors that influences health behaviors. The study hypothesis was that an association exists between family level factors and the caries prevention views and practices of children in Ile-Ife, Nigeria.

## 2. Materials and Methods

### 2.1. Study Design and Study Population

This was a cross-sectional study that recruited study participants from the Ife central local government area of Osun State, a semi-urban town in south-western Nigeria. Participants in the study were children of any gender aged 6 to 11 years old, whose parents agreed to their participation. The exclusion criteria were systemic sickness or learning disability (as reported by parents/guardians). The data were collected through a household survey conducted from December 2018 to January 2019.

### 2.2. Sample Size and Sampling Technique

The minimum sample size for the study was computed using the formula recommended by Araoye [23], based on a caries prevalence of 13.9% [24], a 5% margin of error, and a confidence level of 95%. The calculated sample size was 1233 children.

A multistage sampling strategy was used to recruit the children who participated in the study. Using the simple random sample method, 70 of the 700 enumeration locations in Ife central local government area were selected for the first stage. For the second stage, every other home in the identified enumeration districts was selected. In the last stage, one child from each household who met the inclusion criteria was selected. Households that refused to participate were replaced by the next eligible family. In households with more than one eligible research participant, the children were balloted to choose who would be included in the study. The other children were examined and recommended for treatment

as needed, but their data were not included in the study. Participants were recruited until the calculated sample size was reached.

*2.3. Data Collection*

A questionnaire administered by an interviewer was used to collect data. Field workers were trained on the study procedure, data collecting instruments, sample selection (including household listing and selection), and research ethics [25]. Data obtained comprised independent variables (family structure, family size, childbearing rank), dependent variables (caries preventative views and oral self-care prevention actions), and confounding variables (age at last birthday, sex at birth, socioeconomic status). Data on socioeconomic status were determined using an index proposed by Olusanya et al. [26]. Table 1 provides a summary of the information obtained for the study. Multiple investigations in the study environment had used the index [11,20,27]. When a child had lost a parent, their socioeconomic status was determined by the living parent's status.

We used eight statements based on a five-point Likert scale to assess the children's views on dental caries prevention (Table 1). For each of the eight statements, the responses were scored from one to five with "strongly agree" scoring 5 and "do not know" scoring 1. When no responses were obtained, a score of 1 was assigned. Therefore, each respondent could obtain a total maximum score of 40 and a total minimum score of 8. Using a mean score of 20 as the cut-off point, children who scored 21 or higher were classified as having positive views toward caries prevention, while those who scored 20 or lower were classified as having negative views toward caries prevention. The assessment tool was developed by Khami et al. [28] and had been used in studies conducted in Nigeria [29,30]. The ranking criteria has also been used in prior studies [28,29].

Participants' responses to three questions about the frequency of tooth brushing, the frequency of sugary snacks consumed between main meals, and the usage of fluoridated toothpaste were used to measure comprehensive oral self-care caries prevention practices as shown in Table 1. Brushing more than once daily, consuming sugary snacks between main meals less than once daily, and always using fluoridated toothpaste were the acceptable levels for each component. The number of recommended oral self-care prevention practises adopted by each child was computed and categorized into two parts: those who adopted less than three practises and those who adopted three practises. This methodology was adapted from previous research that assessed the relationship between caries prevention behaviours and the risk of caries in children [31,32].

**Table 1.** Summary information on the study variables.

---

**Independent Variables** *(Possible responses)*

---

1. **Family level influences**
(i) Family size (recoded as family with four or less children and those with more than four children in line with the national policy on family size [33].
(ii) **Family structure (living with both parents, living with mother only, with father only, with mother/father and stepparent or with a caregiver.)**
(iii) **Birth rank** (dichotomized into 'primogenitor (only child) or 'not primogenitor' based on an earlier classification by Ola et al. [20].

---

**Confounding variables** *(Possible responses)*

---

2. **Socio-demographic Characteristics**
(i) **Childs age in years (age at last birthday)**
(ii) **Child's gender (Male or Female)**
(iii) **Family social status using** a multiple-item index [26] combining the mother's level of education with the father's educational level and occupation. **(Grouped for this study as** high (upper and upper-middle classes), middle (middle class) and low (lower- middle and lower classes) socioeconomic status **in line with a prior index regrouping of children in Nigeria** [34].

---

**Table 1.** *Cont.*

---

**Dependent variables.** *(Possible responses)*

---

**1. Caries prevention views**

*(Graded on a 5-point Likert scale "strongly agree" to "agree", "disagree", "strongly disagree", and "do not know")*

**(i)** Fluoridation of drinking water is an effective, safe, and efficient way to prevent dental caries.
**(ii)** Use of fluoride-containing toothpaste is an effective, safe, and efficient way to prevent holes from forming on the teeth.
**(iii)** Frequency of refined carbohydrates consumption has a greater role in producing caries than the total amount of sugar.
**(iv)** Sealant is effective in the prevention of pit and fissure caries in newly erupted molars.
**(v)** Rinsing teeth with a little amount of water after brushing teeth increases the effect of fluoride.
**(vi)** Using fluoride toothpaste is more important than the brushing per se for preventing caries.
**(vii)** Brushing twice daily with fluoride-containing toothpaste is effective for preventing holes from developing in the teeth.
**(viii)** It is important to visit the dental clinic regularly as a measure for preventing holes from forming in the teeth.

---

**2. Caries prevention practices**
**(i)** Frequency of tooth brushing **(Irregularly or never, Once a week, A few (two to three) times a week, once a day, twice a day, more than twice a day)**
**(ii)** Frequency of sugary snacks consumption between main meals **(About three times a day or more, about twice a day, about once a day, occasionally; not every day, rarely or never eat between meals)**
**(iii)** Use of fluoridated toothpaste. **(Always, quite often, seldom, not at all)**

---

## 2.4. Data Analysis

Means and standard deviations for numerical variables were determined, as were frequencies and percentages for categorical variables. To examine the relationship between the dependent and independent variables, bivariate analyses were performed using the chi-squared test. The greatest risk indicator(s) strongly associated with good oral health practises and the adoption of suggested comprehensive oral self-care caries prevention practises were identified using logistic regression analysis. Data analysis was conducted with SPSS for Mac version 28. Statistical significance was set at $p < 0.05$.

## 3. Results

There were 1326 study participants with a mean age (standard deviation) of 8.7 (1.9) years; 600 (45.2%) were female. The majority were graded as low socio-economic status 827 (62.4%), 325 (24.5% 0 were middle social status and 174 (13.1%) were high social status. Of the study participants, 407 (30.7%) had positive caries prevention views, 42 (3.2%) did not use any of the recommended oral self-care prevention practices, and 900 (67.9%), 366 (27.6%), and 18 (1.4%) used one, two, or three recommended oral self-care prevention practices, respectively.

Figure 1 shows the proportion of children who responded appropriately to the questions about caries prevention. The question on the effect of twice daily brushing had the highest number of correct responses (77.9%), while the question on effect of fissure sealants (53.5%) had the lowest number of correct responses.

Figure 2 depicts the proportion of children who reported using the suggested oral self-care behaviours. The majority of the children reported not brushing twice daily (93.0%), always using fluoridated toothpaste (85.4%), and eating sugary meals one or more times per day (73.7%).

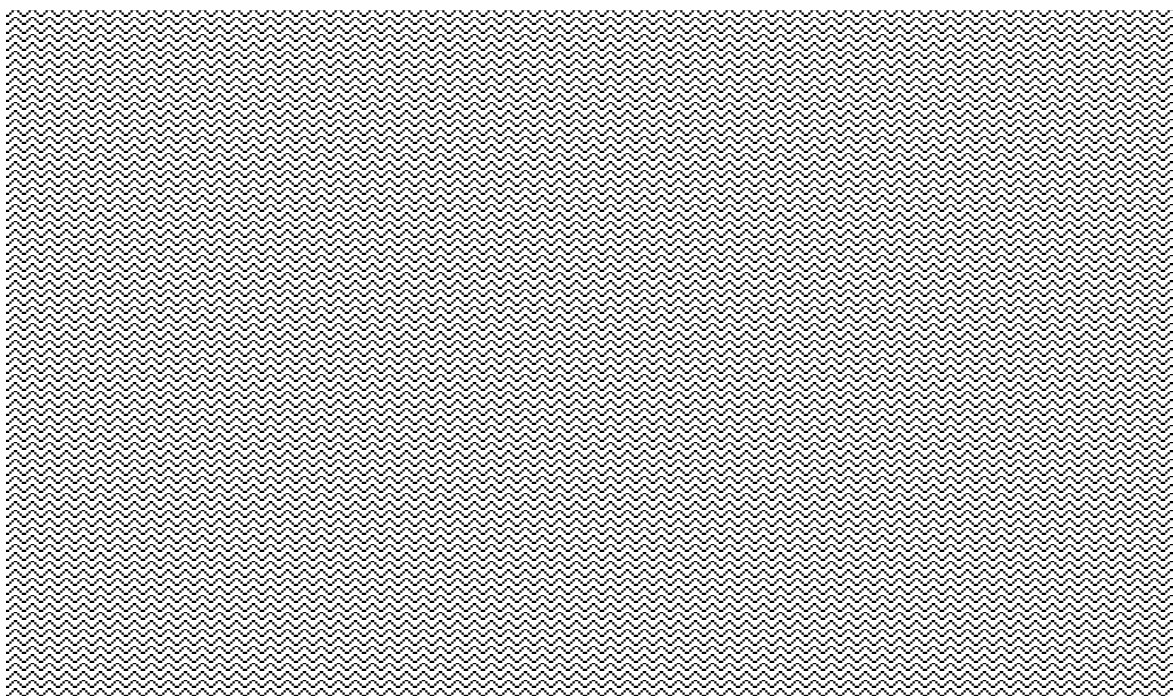

**Figure 1.** Proportion of study participants who responded correctly and incorrectly to the questions assessing caries prevention views.

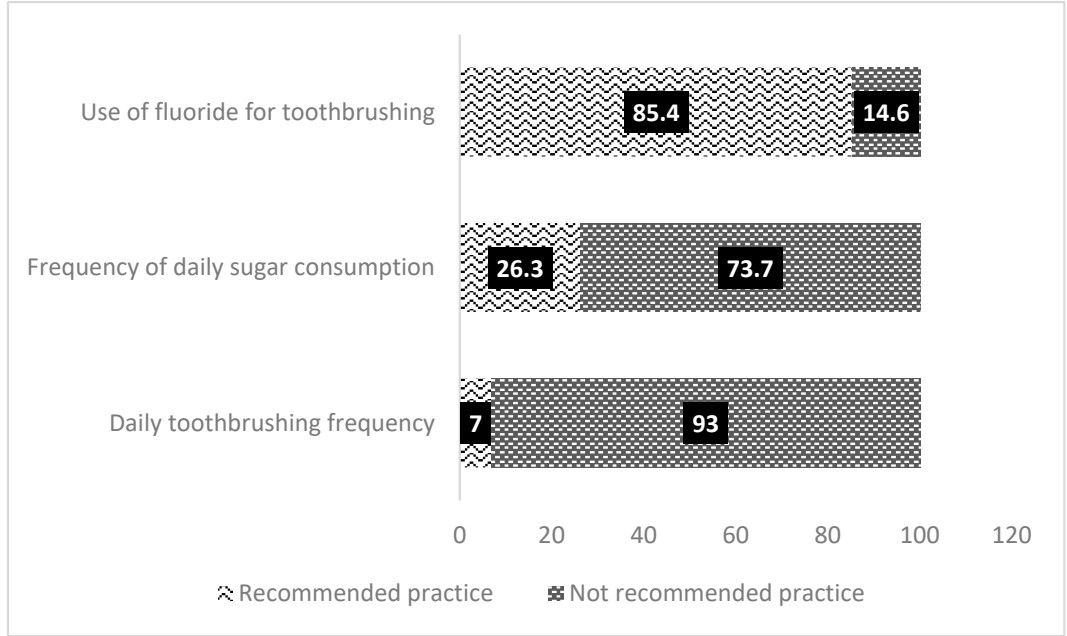

**Figure 2.** Displays the proportion of children who adopted each of the explored oral self-care practices.

Table 2 shows that the proportion of older children who brushed more than once daily was significantly more than the proportion of younger children who brushed more than once daily (*p* = 0.018). Additionally, significantly more children with high social status (*p* = 0.035), from families with more than four children (*p* = 0.002), and who were not progenitor/only child (*p* = 0.033), always/often use fluoride containing toothpaste. In addition, more male than female children consumed refined carbohydrates in-between-meals one or more times daily (*p* = 0.029).

**Table 2.** Association between school-aged children's socio-demographic factors, family level influences, and adoption of recommended oral self-care prevention practices (*N* = 1326).

| Variables | Daily Tooth Brushing Frequency | | Daily Use of Fluoride Containing Toothpaste | | Daily Consumption of Refined Carbohydrates In-Between-Meals | | |
| --- | --- | --- | --- | --- | --- | --- | --- |
| | More Than Once a Day *N* (%) | Once a Day Or Less *N* (%) | Always/Often Use *N* (%) | Seldom Use *N* (%) | Less Than Once a Day *N* (%) | One Or More Times a Day *N* (%) | Total *N* = 1326 *n* (%) |
| **Age** | | | | | | | |
| 6 | 16 (6.3) | 239 (93.7) | 233 (91.4) | 22 (8.6) | 33 (12.9) | 222 (87.1) | 255 (19.2) |
| 7 | 12 (5.2) | 219 (94.8) | 1215 (93.1) | 16 (3.2) | 29 (12.6) | 202 (87.4) | 231 (17.4) |
| 8 | 13 (5.2) | 235 (94.8) | 240 (96.8) | 8 (4.6) | 30 (12.1) | 218 (87.9) | 248 (18.7) |
| 9 | 17 (7.2) | 220 (92.8) | 226 (95.4) | 11 (13.1) | 25 (10.5) | 212 (89.5) | 237 (17.9) |
| 10 | 14 (6.9) | 190 (93.1) | 187 (91.7) | 17 (8.3) | 27 (13.2) | 177 (86.8) | 204 (15.4) |
| 11 | 21 (13.9) | 130 (86.1) | 143 (94.7) | 8 (5.3) | 12 (7.9) | 139 (92.1) | 151 (11.4) |
| *p*-value | 0.018 * | | 0.095 | | 0.640 | | |
| **Sex** | | | | | | | |
| Female | 51 (8.5) | 549 (91.5) | 523 (87.2) | 77 (12.8) | 140 (23.3) | 460 (76.7) | 600 (45.2) |
| Male | 42 (5.8) | 684 (94.2) | 610 (84.0) | 116 (16.0) | 209 (28.2) | 517 (71.8) | 726 (54.8) |
| *p*-value | 0.69 | | 0.106 | | 0.029 * | | |
| **Social status** | | | | | | | |
| Low | 56 (6.8) | 771 (93.2) | 712 (86.1) | 115 (13.9) | 221 (26.7) | 606 (73.3) | 827 (62.4) |
| Middle | 27 (8.3) | 298 (91.7) | 265 (81.5) | 60 (18.5) | 83 (25.5) | 242 (74.5) | 325 (25.5) |
| High | 10 (5.7) | 164 (94.3) | 156 (89.7) | 18 (10.3) | 45 (25.9) | 129 (74.1) | 174 (13.1) |
| *p*-value | 0.513 | | 0.035 * | | 0.909 | | |
| **Family size** | | | | | | | |
| 4 or less children | 84 (7.2) | 1081 (92.8) | 983 (84.4) | 182 (15.6) | 308 (23.2) | 857 (73.6) | 1165 (87.8) |
| More than 4 children | 9 (5.6) | 152 (94.4) | 150 (93.2) | 11 (6.8) | 41 (25.5) | 120 (74.5) | 161 (12.2) |
| *p*-value | 0.514 | | 0.002 * | | 0.867 | | |
| **Family structure** | | | | | | | |
| Living with both parents | 82 (6.8) | 1129 (93.2) | 1036 (85.5) | 175 (14.5) | 317 (26.2) | 894 (73.8) | 1211 (91.3) |
| All other living arrangements | 11 (9.6) | 104 (90.4 | 97 (84.3) | 18 (15.7) | 32 (27.8) | 83 (72.2) | 115 (8.7) |
| *p*-value | 0.262 | | 0.833 | | 0.701 | | |
| **Child's birth rank** | | | | | | | |
| Progenitor or only child | 40 (8.2) | 450 (91.8) | 405 (82.7) | 85 (17.3) | 137 (28.0) | 353 (72.0) | 490 (37.0) |
| Not progenitor or only child | 53 (6.3) | 783 (93.7) | 728 (87.1) | 108 (12.9) | 212 (25.4) | 624 (74.6) | 836 (63.0) |
| *p*-value | 0.253 | | 0.033 * | | 0.330 | | |
| Total | 93 (7.0) | 1233 (93.0) | 1133 (85.4) | 193 (14.6) | 349 (26.3) | 977 (73.7) | |

\*—statistically significant.

As shown in Table 3, the model assessing the factors significantly associated with positive caries prevention views was statistically significant (*p* = 0.006). While the model assessing the factors significantly associated with the use of recommended oral self-care prevention practices by children, was not statistically significant. The models explained only a very small proportion of the results obtained (caries prevention knowledge = 2.0%;

adoption of recommended prevention practices = 6.3%). Children from larger families had significantly lower odds of positive caries prevention views compared to those from smaller families (AOR: 0.572; 95% CI: 0.404–0.809; $p$ = 0.002). The child's family structure was significantly associated with the use of recommended oral self-care prevention practices. Children not living with both parents had significantly increased odds of adopting recommended oral self-care caries prevention methods (AOR: 3.165; 95% CI: 1.011–9.902; $p$ = 0.048).

**Table 3.** Association between children's socio-demographic indicators caries prevention views and use of recommended oral self-care caries prevention practices by 6–11-year-old children in Nigeria ($N$ = 1326).

| Variables | Caries Prevention Views AOR (95% CI) | *p*-Value | Recommended Self-Care Caries Prevention Practices AOR (95% CI) | *p*-Value |
| --- | --- | --- | --- | --- |
| **Family size** | | | | |
| Four or less children in the family | 1.000 | - | 1.000 | - |
| More than 4 children | 0.572 (0.404–0.809) | 0.002 * | 0.354 (0.045–2.759) | 0.321 |
| **Family structure** | | | | |
| Living with both parents | 1.000 | - | 1.000 | - |
| All other living arrangements | 0.770 (0.520–1.139) | 0.191 | 3.165 (1.011–9.902) | 0.048 * |
| **Child's birth position** | | | | |
| First or only child | 1.000 | - | 1.000 | - |
| Not first or only child | 1.089 (0.853–1.389) | 0.494 | 1.054 (0.397–2.797) | 0.916 |
| **Nagelkerke R Square** | 0.020 | - | 0.063 | - |
| **-2Log Likelihood** | 1707.847 | 0.006 * | 179.246 | 0.222 |

*—statistically significant.

## 4. Discussion

Understanding the factors that influence oral health-related views among children is critical for developing effective and efficient caries prevention measures especially in low-income countries such as Nigeria where access to oral health care is limited. Children's oral health views can influence their oral health practices, such as proper teeth brushing technique, brushing time, brushing frequency, usage of dental floss, and cleaning after meals [35]. The development of positive views and practises can play a major part in creating good attitudes to oral health that can be lifelong [36]. The findings from the current study shows that the proportion of the study participants who have positive views about caries prevention views was low. The family-level factor associated with caries prevention views was family size: children from smaller households seem more likely to have positive views about caries prevention. On the contrary, children living with both parents were less likely to use oral self-care caries preventative techniques. The study results only partially support the study hypothesis.

One of the strengths of this study is the methodology. A household survey was conducted which makes the study findings generalizable for the community. This is also the first study in Nigeria to assess the family level factors that may improve the design and implementation of oral health education programs for school-aged children. The cross-sectional study design, however, limits our ability to make causal inferences from the study findings. It also limits the ability to determine the temporal link between the dependent and independent variables because both were studied at the same time. Furthermore, there was also the risk for social desirability reporting of the use of caries preventive measures though this may have been reduced to a bare minimum through the training of the field workers

who learnt how to ask questions in disarming ways. The high number of respondents who reported non-use of the caries prevention tools may be an indication of the low risk of social desirability responses for this study. In addition, we have very low correct response on the question on fissure sealant, which may be a reflection on poor understanding about this material for oral health care by school-age children. Despite these study limitations, our results present several important findings.

First, the only family-level factor associated with the child's caries prevention views was the family size. Smaller size families reflect higher socioeconomic status [37] as indicated in prior studies conducted among children in Nigeria [10,11]. Children from high socio-economic status families are more likely to have access to information and education about oral health [38]. The study result is, therefore, not surprising. However, the observed association between positive caries prevention views and socioeconomic status at bivariate level was not replicated when it came to caries prevention practices—the use of recommended oral self-care caries preventive practices was not significantly associated with the socioeconomic status. This study finding may indicate that though the child's view of caries prevention is important, it may not be enough to shape or influence oral health related behavior in school-age children.

We also observed that children living with both parents were less likely to use oral self-care caries preventative techniques. A similar pattern was reported in a multinational study involving 32 nations on toothbrushing and family characteristics [39]. Only a few nations had a higher prevalence of children living in two-parent families reporting adoption of the recommended toothbrushing frequency. This study suggests that the finding by Simon [40] that divorced/single parents are less likely than two-parent families to monitor and discipline their children about toothbrushing may not be valid in all countries. This finding warrants more investigation in order to better understand the context specific factors that influence children's toothbrushing habits.

Most school-aged children are dependent on their parents [41]. Parental attributes, especially maternal attributes may, therefore, be powerful in shaping the oral health behaviour of school-aged children. This could explain why the logistic regression model for this study explains just a small fraction of the family factors that influence positive preventive views and the use of suggested oral self-care caries prevention behaviours by school-age children. Including parenting attributes may have strengthened this model as prior studies had suggested this association [42,43]. There are, however, no prior studies conducted in the study population that suggests a link between parenting and caries risk. Future studies on how parenting influence the caries risk profile of school-aged children in Nigeria may be helpful. Culture modulates the associations between parenting and children's health risk [44]. For this reason, results from other cultures cannot be extrapolated to Nigeria.

Second, the proportion of children with positive caries prevention views was low, even though the results suggest that the children's opinions improved with age. The results also seem to indicate that though the proportion of children who practiced recommended oral self-care caries prevention was low, the practice seems to increase with age. The suboptimal views about caries prevention and inadequate practice of suggested oral self-care caries prevention may increase the risk for both caries and gingivitis. Ironically, the prevalence of caries is low in the study population [5], despite the high proportion of children who consume refined carbohydrate in-between-meals daily [29]. This result could be attributed to the high proportion of children who use fluoride toothpaste. Further studies are recommended to fully understand the current study's results, which may inform the design of intervention programs to increase school children's uptake of caries prevention strategies.

The findings of the current study suggest complex interactions between risk factors, oral disease outcomes, and the possible mediating/moderating role(s) of caries prevention views. The family contexts and culture may influence these interactions. Oral health promotion interventions may address the highlighted gaps in the caries prevention views of

children in the study community. In addition, the national policy that promotes small family sizes of four children may have a positive impact on children's oral health. Oral health programs in Nigeria that promote caries prevention may be strengthened by advocating for synergies with family planning policies as this can help with the design of cost-effective integrated programs that may increase the uptake of both contraceptives and oral health services in Nigeria. Uptake of both services are currently low in Nigeria. Further studies are needed to identify other family-level related caries protective factors for children aged 6–11, who are still highly dependent on parental care and guidance.

## 5. Conclusions

Family size and living arrangement may influence school children's caries prevention views and practices. School children from larger families in developing countries may benefit from targeted oral health promotion activities. The public debate on birth control and family size control should include comments on the positive impact of family size control on oral health.

**Author Contributions:** Conceptualization, A.A.A. and M.O.F.; methodology, M.O.F., O.A. and M.E.T.; validation, A.A.A., M.E.T. and M.O.F.; formal analysis, A.A.A.; investigation, N.M.C.; writing—original draft preparation, A.A.A. and M.O.F.; writing—review and editing, A.A.A., M.O.F., O.A., N.M.C. and M.E.T.; supervision, M.O.F. All authors have read and agreed to the published version of the manuscript.

**Funding:** This research received no external funding.

**Institutional Review Board Statement:** The study was conducted in accordance with the Declaration of Helsinki and approved by the I Ethics and Research Committee of the Institute of Public Health, Obafemi Awolowo University, Ile-Ife, Nigeria (IPH/OAU/12/1887) Ethics and Research Committee of the Institute of Public Health, Obafemi Awolowo University, Ile-Ife, Nigeria (IPH/OAU/12/1887).

**Informed Consent Statement:** Informed consent was obtained from all subjects and their parents involved in the study.

**Data Availability Statement:** The data presented in this study are available on request from the corresponding author. The data are not publicly available due to ethical restrictions.

**Conflicts of Interest:** The authors declare no conflict of interest.

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
