# Peer review of "Association between Family Level Influences and Caries Prevention Views and Practices of School Children in a Sub-Urban Nigerian Community"

_2673-8430, doi:10.3390/biomed3010011_

Round 1

Reviewer 1 Report

A revision of manuscript's english language is necessary before resubmission.

All text should be justified.

The introduction could be better argued.

Line 52 should be removed.

Line 61 should be removed.

The discussion and conclusion section should be revised. There are several repeated concepts. Furthermore, I suggest to the authors to add the limitation of the study and provide the suggestions and future perspectives related to the findings in a more detailed way. 

The reference section should be improved with more recent studies.

Line 266- typo should be corrected.

Author Response

Dear Reviewer,

Thank you for reading our work and for your thoughtful comments. We have responded to the concerns and suggestions highlighted in red below. We've also updated the manuscript text to reflect the changes and highlighted the revised areas in red. We believe that the modified paper flows better, is more informative, and makes a valuable contribution to the topic of the association between family level factors and caries prevention views and practices of school Nigerian children

A revision of manuscript's English language is necessary before resubmission.

R: Thank you for the feedback We have revised the manuscript extensively. The edited areas are highlighted in red in the revised manuscript.

All text should be justified.

R: The text has been justified.

The introduction could be better argued.

R: We have included more information in the introduction. We have also reviewed the justification for the study. We wrote that “Considering the levels indicated in Fisher-Owen’s framework, it may be necessary to consider family level interventions in addition to community level interventions for promoting the adoption of preventive oral health practises and increasing access to oral health in Nigerian school children. Furthermore, evidence suggests that programmes that target various levels of factors influencing oral health in children are more effective than those that target only the individual level [22]. This is especially crucial when promoting equitable access to oral health care for children.”

Line 52 should be removed.

R: Line 52 has been removed.

Line 61 should be removed.

R: Line 61 has been removed.

The discussion and conclusion section should be revised. There are several repeated concepts. Furthermore, I suggest to the authors to add the limitation of the study and provide the suggestions and future perspectives related to the findings in a more detailed way. 

R: We have revised the discussion and conclusion section extensively as suggested

The reference section should be improved with more recent studies.

 R: We have included a few more recent references in the manuscript.

Line 266- typo should be corrected.

R: The typo has been corrected.

Reviewer 2 Report

The manuscript under review attempts to evaluate the association between family level influences, and caries prevention views and practices of school children in a sub-urban Nigerian community.  In general, the manuscript captures details of the study design and implementation of the project. All the sections of the manuscript are well written and concluded, although the results are less, and it has been presented in the manuscript. The study is of sound design and of clear practical and clinical interest, I suggest accepting this article with major revision.

Abstract:

1.     Provide structured abstract.

2.     Kindly mention the significant values

3.     Provide Mesh keywords.

Introduction:

1.     Kindly write the justification of the study

2.     Write about school dental health programs.

3.      Kindly write about family level factors with literature

4.     Fisher-Owens child oral health conceptual framework: provide details if possible illustrate.

M and M:

1.     Kindly write inclusion criteria

2.     From line 112 to 152 can be presented as table. Kindly present it as tables and describe it briefly.

3.     Data analysis: kindly mention the software used.

Results:

1.     Results are less, but well presented.

2.     Kindly cite the significant difference in the tables

Discussion:

1.     Discuss the importance of caries prevention views and practices with literature.

2.     Write limitation of the study

3.     How will the outcome help the community?

4.     Preventive care implementations

5.     Recommendations

Author Response

Dear Reviewer,

Thank you for reading our work and for your thoughtful comments. We have responded to the concerns and suggestions highlighted in red below. We've also updated the manuscript text to reflect the changes and highlighted the revised areas in red. We believe that the modified paper flows better, is more informative, and makes a valuable contribution to the topic of the association between family level factors and caries prevention views and practices of school Nigerian children.

  1. Kindly write the justification of the study

R: The study justification has been revised and strengthened. We wrote “Considering the levels indicated in Fisher-Owen’s framework, it may be necessary to consider family level interventions in addition to community level interventions for promoting the adoption of preventive oral health practises and increasing access to oral health in Nigerian school children. Furthermore, evidence suggests that programmes that target various levels of factors influencing oral health in children are more effective than those that target only the individual level [22]. This is especially crucial when promoting equitable access to oral health care for children.”

  1. Write about school dental health programs. 

R: Information on school oral health programs have been included. We wrote “School-based oral health programmes are the most popular programme approach for addressing preventive oral health in school-aged children, as they have the potential to promote equitable access to oral health care. It is also a strategy supported by the World Health Organization's oral health programme [21]. Schools are considered ideal setting for improving oral health because they provide an efficient and effective approach to reach children and, through them, families, and community members. Addressing oral health in childhood, also can establish lifelong, sustainable oral health related behaviours, beliefs, and attitudes.”

  1. Kindly write about family level factors with literature 

R: We have included information on family level factors in the text as “Family-level influences on children’s oral health behavior are predominantly driven by mothers [13, 14], although older siblings reportedly wield some influence [15]. Research further indicates children’s caries experience is significantly worse when family functioning is poor [16, 17] and parenting styles are authoritative or permissive [18]. Reported family level influences on the caries risk of children resident in Nigeria include birth rank [19], family size [20], and family structure [20].”

  1. Fisher-Owens child oral health conceptual framework: provide details if possible, illustrate. 

R: We have included a description of the Fisher-Owens model in the manuscript. “Families influence children's oral health both directly and indirectly by giving support and role modelling, as indicated in the Fisher-Owens child oral health conceptual model [12]. The Fisher-Owens model is a multilevel conceptual framework that recognizes the complex interactions between individual, family-level, and community-level factors in any of the five domains of health determinants, one of which is health behaviors. It also recognizes the impact of the complex interactions between the model's various levels on the child's oral health [12].”

M and M:

  1. Kindly write inclusion criteria 

R: We have included the inclusion criteria: “The following inclusion criteria were used: any gender, age should be six to eleven years, and the presence of a parent/guardian fluent in English or the local language.”

  1. From line 112 to 152 can be presented as table. Kindly present it as tables and describe it briefly. 

R: We have inserted a Table to provide a summary of the independent, dependent, and confounding variables.

  1. Data analysis: kindly mention the software used.

R: We used SPSS version 28 for Mac for the data analysis. This has been indicated in the manuscript.

Results:

  1. Results are less, but well presented. 

R: Thank you for the feedback. We have included a figure showing the participants reported oral-self-care methods.

  1. Kindly cite the significant difference in the tables 

R: The significant results un the results have been cited in the text. In the data analysis for this study, the child's age, gender, and social standing were considered confounding variables, hence the association with the outcome variable in the regression model should not be reported. The revised manuscript corrects this.

Discussion:

  1. Discuss the importance of caries prevention views and practices with literature. 

R: We have included information of caries prevention views. We wrote that “Understanding the factors that influence oral health-related views among children is critical for developing effective and efficient caries prevention measures especially in low-income countries like Nigeria where access to oral health care is limited. Children's oral health views can influence their oral health practices such as proper teeth brushing technique, brushing time, brushing frequency, usage of dental floss, and cleaning after meals [35]. The development of positive views and practises can play a major part in creating good attitudes to oral health that can be lifelong [36].”

  1. Write limitation of the study

R: The study limitations have been further revised: we wrote that “The cross-sectional study design however, limits our ability to make causal inferences from the study findings. It also limits the ability to determine the temporal link between the dependent and independent variables because both were studied at the same time. Furthermore, there was also the risk for social desirability reporting of the use of caries preventive measures though this may have been reduced to a bare minimum through the training of the field workers who learnt how to ask questions in disarming ways.”

  1. How will the outcome help the community? 

R: We included the following statement in the manuscript: “Oral health promotion interventions may address the highlighted gaps in the caries prevention views of children in the study community. In addition, the national policy that promotes small family sizes of four children may have a positive impact on children's oral health. Oral health programs in Nigeria that promote caries prevention may be strengthened by advocating for synergies in family planning policies. Highlighting the link between family planning and children's oral health could be used to boost family planning programmes in Nigeria and promote oral health. Further studies are needed to identify other family-level related caries protective factors for children aged 6 to 11, who are still highly dependent on parental care and guidance.”

  1. Preventive care implementations

R: We have included the following in the manuscript “Oral health promotion interventions may address the highlighted gaps in the caries prevention views of children in the study community. In addition, the national policy that promotes small family sizes of four children may have a positive impact on children's oral health. Oral health programs in Nigeria that promote caries prevention may be strengthened by advocating for synergies in family planning policies.”

  1. Recommendations

R: Based on the study findings, our recommendation include: “Further studies are recommended to fully understand the current study’s results, which may inform the design of intervention programs to increase schoolchildren’s uptake of caries prevention strategies.”

We also wrote that “Oral health programs in Nigeria that promote caries prevention may be strengthened by advocating for synergies in family planning policies. Highlighting the link between family planning and children's oral health could be used to boost family planning programmes in Nigeria and promote oral health. Further studies are needed to identify other family-level related caries protective factors for children aged 6 to 11, who are still highly dependent on parental care and guidance.”  

In the conclusion we noted that “The public debate on birth control and family size control should include comments regarding the favourable influence on oral health.”

Round 2

Reviewer 1 Report

Requests have been met. Manuscript has been improved.

Reviewer 2 Report

Dear Authors,

The authors have addressed all the comments and manuscript much improved. I would like to congratulate the authors and wish them all the very best for future endeavour